# Convenient Synthesis of Pyrazolo[4′,3′:5,6]pyrano[4,3-*c*][1,2]oxazoles via Intramolecular Nitrile Oxide Cycloaddition

**DOI:** 10.3390/molecules26185604

**Published:** 2021-09-15

**Authors:** Vaida Milišiūnaitė, Elena Plytninkienė, Roberta Bakšienė, Aurimas Bieliauskas, Sonata Krikštolaitytė, Greta Račkauskienė, Eglė Arbačiauskienė, Algirdas Šačkus

**Affiliations:** 1Institute of Synthetic Chemistry, Kaunas University of Technology, K. Baršausko g. 59, LT-51423 Kaunas, Lithuania; vaida.milisiunaite@ktu.lt (V.M.); elena.plytninkiene@ktu.lt (E.P.); aurimas.bieliauskas@ktu.lt (A.B.); greta.ragaite@ktu.lt (G.R.); 2Department of Organic Chemistry, Kaunas University of Technology, Radvilėnų pl. 19, LT-50254 Kaunas, Lithuania; roberta.ramanauskaitee@gmail.com (R.B.); sonata.krikstolaityte@ktu.lt (S.K.)

**Keywords:** pyrazole, isoxazoline/isoxazole, fused ring systems, intramolecular nitrile oxide cycloaddition, 4-pyrazolaldoximes ^1^*J*_CH_, isoxazole ^15^N NMR, isoxazoline ^15^N NMR

## Abstract

A simple and efficient synthetic route to the novel 3a,4-dihydro-3*H*,7*H*- and 4*H*,7*H*-pyrazolo[4′,3′:5,6]pyrano[4,3-*c*][1,2]oxazole ring systems from 3-(prop-2-en-1-yloxy)- or 3-(prop-2-yn-1-yloxy)-1*H*-pyrazole-4-carbaldehyde oximes has been developed by employing the intramolecular nitrile oxide cycloaddition (INOC) reaction as the key step. The configuration of intermediate aldoximes was unambiguously determined using NOESY experimental data and comparison of the magnitudes of ^1^*J*_CH_ coupling constants of the iminyl moiety, which were greater by approximately 13 Hz for the predominant *syn* isomer. The structures of the obtained heterocyclic products were confirmed by detailed ^1^H, ^13^C and ^15^N NMR spectroscopic experiments and HRMS measurements.

## 1. Introduction

The 1,3-dipolar cycloaddition reaction of nitrile oxides as 1,3-dipoles and alkenes/alkynes as dipolarophiles has become an efficient tool in organic synthesis to obtain various substituted isoxazolines/isoxazoles [1,2,3]. The reaction was developed by Rolf Huisgen and described by Albert Padwa in their investigations on 1,3-dipolar cycloadditions [4,5]. Nitrile oxides, which are typically generated in situ, undergo subsequent 1,3-dipolar cycloaddition to form appropriate isoxazoles or isoxazolines. Numerous methods of nitrile oxide generation have been reported, mainly including the dehydration of nitroalkanes [6,7,8] and oxidation of aldoximes [9,10,11]. Alternatively, Svejstrupor described the synthesis of isoxazolines and isoxazoles from hydroxyimino acids via the visible-light-mediated generation of nitrile oxides by two sequential oxidative single electron transfer processes [12]. More recently, Chen et al. reported the synthesis of fully substituted isoxazoles from nitrile oxides, which were generated in situ from copper carbene and *tert*-butyl nitrite [13].

Notably, the intramolecular nitrile oxide cycloaddition (INOC) reaction can provide a route for the preparation of isoxazoles or isoxazolines annulated to various carbo- or heterocycles. For example, the intramolecular 1,3-dipolar cycloaddition of 2-phenoxybenzonitrile *N*-oxides to neighboring benzene rings, accompanied by dearomatization, formed the corresponding isoxazolines in high yields [14]. Recently, a method for the stereoselective synthesis of novel isoxazoline/isoxazole-fused indolizidine-, pyrrolizidine- and quinolizidine-based iminosugars has been developed, employing *N*-alkenyl/alkynyl iminosugar *C*-nitromethyl glycosides as nitrile oxide precursors in 1,3-dipolar cycloaddition reactions [15]. The phthalate-tethered INOC strategy has also been described as a novel method for the synthesis of 12–15-membered chiral macrocycles having a bridged isoxazoline moiety in a highly regio- and diastereoselective manner [16]. Furthermore, diversity-oriented access to isoxazolino and isoxazolo benzazepines as possible bromodomain and extra-terminal motif protein (BET) inhibitors has been reported via a post-Ugi heteroannulation involving the intramolecular 1,3-dipolar cycloaddition reaction of nitrile oxides with alkenes and alkynes [17]. In addition, an intramolecular 1,3-dipolar nitrile oxide cycloaddition strategy has been applied as an efficient synthesis protocol for the regio- and diastereoselective construction of highly functionalized tricyclic tetrahydroisoxazoloquinolines [18].

Fused isoxazoles or isoxazolines obtained by the INOC reaction may also serve as synthetically important intermediates for many biologically active compounds. Such compounds, including the HBV inhibitor entecavir [19,20], the antibiotic branimycin [21], the antiviral (+)-Brefeldin A [22], tricyclic isoxazoles combining serotonin (5-HT) reuptake inhibition with α_2_-adrenoceptor blocking activity [23] and the alkaloids meliacarpinin B [24] and Palhinine A [25], have been synthesized by employing INOC as a key step.

We previously investigated the metal-free intramolecular alkyne-azide cycloaddition reaction for the formation of the pyrazolo[4,3-*f*][1,2,3]triazolo[5,1-*c*][1,4]oxazepine ring system [26] as well as the synthesis of 2,6-dihydropyrano[2,3-*c*]pyrazole derivatives by employing the ring-closing metathesis (RCM) reaction [27]. In continuation of our interest in the synthesis and investigation of novel pyrazole-containing polyheterocyclic systems [28,29,30,31,32,33], we report herein the synthesis and structural elucidation of new 3a,4-dihydro-3*H*,7*H*- and 4*H*,7*H*-pyrazolo[4′,3′:5,6]pyrano[4,3-*c*][1,2]oxazole derivatives from appropriate 3-(prop-2-en-1-yloxy)- or 3-(prop-2-yn-1-yloxy)-1*H*-pyrazole-4-carbaldehyde oximes via the intramolecular nitrile oxide cycloaddition reaction.

## 2. Results

The synthetic strategy that we designed to construct the pyrazolo[4′,3′:5,6]pyrano[4,3-*c*][1,2]oxazole ring system employs difunctional substrates (**4a**–**d**) that contain an aldoxime unit next to the allyloxy group attached to the pyrazole core and can serve as intermediates for nitrile oxide generation and subsequent cycloaddition (Figure 1).

As starting materials for the synthesis of compounds **4a**–**d**, we used 1-phenyl-, 1-(4-fluorophenyl)-, 1-(4-bromophenyl)- and 1-methylpyrazol-3-ols (**1a**–**d**), which are readily accessible from the oxidation of appropriate pyrazolidin-3-ones [34]. The *O*-allylation of **1a**–**d** with allylbromide in the presence of NaH gave *O*-allylated pyrazoles **2a**–**d** [27]. To introduce a formyl group to the 4-position of the pyrazole ring, we employed a previously reported Vilsmeier–Haack reaction procedure [27,35]. Heating compounds **2a**–**d** with Vilsmeier–Haack complex at 70 °C resulted in the formation of the desired pyrazole-4-carbaldehydes **3a**–**d** (Figure 1).

In order to prepare the 3a,4-dihydro-3*H*,7*H*-pyrazolo[4′,3′:5,6]pyrano[4,3-*c*][1,2]oxazole derivatives **5a**–**d** by the INOC reaction, aldoximes **4a**–**d** were synthesized by the treatment of **3a**–**d** with hydroxylamine hydrochloride in the presence of sodium acetate [36]. As a result, the *syn-* and *anti*-3-allyloxy-4-pyrazolaldoximes were obtained in total yields of 82–97%. The ^1^H NMR spectra of aldoximes **4a**–**c** showed the presence of two isomers in different ratios with a predominance of the *syn* isomer, while the compound **4d** was obtained as a pure *syn* isomer.

Over the years, the isomerism of aldoximes has been thoroughly studied and many different NMR-based approaches have been developed, mainly due to the large differences in chemical shifts, coupling constants and distinct through-space connectivities in NOESY measurements of aldoxime *syn*-*anti* isomers [37]. The configurational assignment of aldoximes **4a**–**c** was relatively easy due to the presence of both isomers, as it is well established that the resonance of the iminyl-H proton in the *syn* isomer is greatly shifted upfield by approximately δ 0.5–0.7 ppm in the ^1^H NMR spectra compared to the *anti* isomer [38]. Moreover, the 1D selective NOESY experimental data of aldoximes **4a**–**c** showed that, upon irradiation of the hydroxyl proton N-OH of the predominant *syn* isomer, a strong positive NOE on the pyrazole 5-H proton was observed, while the minor isomer showed a positive NOE on the iminyl-H, therefore confirming the *anti* configuration. Finally, a heteronuclear 2D *J*-resolved NMR experiment was used in order to determine ^1^*J*_CH_ coupling constants throughout the series of aldoximes. It is well established from previous studies that there is a large and constant difference between the magnitudes of ^1^*J*_CH_ coupling constants of the iminyl moiety in *syn*-*anti* isomers [39], which is larger by at least 10–15 Hz for the *syn* isomer. The measurements of compounds **4a**–**c** showed that the relevant ^1^*J*_CH_ coupling constants of the iminyl moiety were around 175.0 Hz for the predominant *syn* isomer, while the minor *anti* isomer provided significantly lower coupling constant values by around 13.0 Hz. The configuration of aldoxime **4d** as a pure *syn* isomer was easily deduced from NOESY measurements and the ^1^*J*_CH_ coupling constant of the iminyl moiety, which was 174.5 Hz. The analysis of ^15^N NMR spectroscopic data showed highly consistent chemical shift values within each isomer, in a range from δ −18.2 to −25.7 ppm in the case of the *syn* isomer and in a range from δ −15.6 to −16.5 ppm for the *anti* isomer. A comparison of the relevant NMR data of aldoximes is presented in Appendix A.

Several methods for the oxidation of aldoximes to nitrile oxides are known in the literature, including the application of oxidants such as chloramine T [40,41], *N*-halosuccinimides (NXS) [42,43,44], hypohalites [45,46,47], hypervalent iodine reagents [48,49,50] and oxone [51,52,53,54]. The reaction conditions for 3a,4-dihydro-3*H*,7*H*-pyrazolo[4′,3′:5,6]pyrano[4,3-*c*][1,2]oxazole ring formation were optimized by using **4a** as a model compound (Table 1). When treating aldoxime **4a** with chloramine T in EtOH at 50 °C for 30 min, the polycyclic product **5a** was obtained in poor (20%) yield (Table 1, Entry 1). The intramolecular cyclization reaction of **4a** in the presence of the aq. NaOCl in DCM gave the desired product **5a** in 1 h in sufficient (68%) yield (Table 1, Entry 2). The experiment with TEA as an additive did not improve the yield of the product and **5a** was obtained in 52% yield (Table 1, Entry 3). A similar result showing that no additional base is required to facilitate the cycloaddition was also observed by Roy and De in their investigation on the rate enhancement of nitrile oxide cyclization and, hence, rapid synthesis of isoxazolines and isoxazoles [55].

The optimized conditions (aq. NaOCl in DCM at rt) for **5a** synthesis were also applied to the synthesis of 7-(4-fluorophenyl)-, 7-(4-bromophenyl)- and 7-methyl-3a,4-dihydro-3*H*,7*H*-pyrazolo[4′,3′:5,6]pyrano[4,3-*c*][1,2]oxazoles **5b**–**d** to evaluate the scope of the methodology. The products were obtained in yields of 63%, 64% and 42%, respectively. In addition, we investigated whether the obtained 3a,4-dihydro-3*H*,7*H*-pyrazolo[4′,3′:5,6]pyrano[4,3-*c*][1,2]oxazole system can be further oxidized. Several oxidation reaction conditions were tested, e.g., **5a** was stirred in DMSO at 110 °C in an open atmosphere [56] or treated with a catalytic amount of Pd/C in acetic acid [57]; the best result was obtained using activated MnO_2_ as an oxidant in toluene in a Dean–Stark apparatus for 4 h at reflux temperature [58]. Furthermore, 4*H*,7*H*-Pyrazolo[4′,3′:5,6]pyrano[4,3-*c*][1,2]oxazole derivative **6** was formed in 38% yield.

A similar brief study on 5-chloropyrazole-4-carbaldehydes as synthons for intramolecular 1,3-dipolar cycloaddition was also reported by L‘abbé et al. [59]. The authors noticed that 5-allyloxypyrazole-4-carbaldehyde derived from 5-chloropyrazole-4-carbaldehyde and further used as a precursor for intramolecular 1,3-dipolar cycloaddition reactions underwent a slow Claisen rearrangement to 4-allyl-5-hydroxypyrazole, even at room temperature. In contrast, we found 3-allyloxypyrazole-4-carbaldehydes to be stable. They can be stored in the laboratory at room temperature.

The formation of 3a,4-dihydro-3*H*,7*H*- and 4*H*,7*H*-pyrazolo[4′,3′:5,6]pyrano[4,3-*c*][1,2]oxazole ring systems was easily deduced after an in-depth analysis of NMR spectral data, which were obtained through a combination of standard and advanced NMR spectroscopy techniques, such as ^1^H-^13^C HMBC, ^1^H-^13^C *J*-HMBC, ^1^H-^15^N HMBC, ^1^H-^13^C HSQC, ^1^H-^13^C H2BC, ^1^H-^1^H COSY, ^1^H-^1^H NOESY and 1,1-ADEQUATE experiments (Figure 1).

In the case of compound **5a**, the multiplicity-edited ^1^H-^13^C HSQC spectrum allowed us to identify the pairs of geminally coupled methylene protons, since both protons displayed cross-peaks with the same carbon. For instance, it showed two pairs of negative signals at δ_H_ 4.66, 3.79 and 4.78, 4.17 ppm, which have one-bond connectivities with the methylene carbons C-3 (δ 69.7 ppm) and C-4 (δ 70.9 ppm), respectively. The chemical shifts of these methylene groups are expected to be similar and downfield compared to a neighboring methine group at site 3a, because both are bound to the oxygen atoms O-2 and O-5. This adjacent protonated carbon C-3a (δ 46.7 ppm) relative to the aforementioned methylene sites was easily assigned from an appropriate correlation in the ^1^H-^13^C H2BC spectrum.

In the ^1^H-^15^N HMBC spectrum of **5a**, strong long-range correlations between the methylene 3-H proton at δ 4.66 ppm and the 3a-H proton at δ 3.86–3.91 ppm with the oxazole N-1 nitrogen at δ −32.2 ppm were observed. The lack of long-range correlations with another pair of methylene protons (δ 4.78, 4.17 ppm), and the aforementioned N-1 nitrogen, strongly hinted at assigning this methylene group to site 4. In order to unambiguously discriminate between these methylene groups, the ^1^H-^13^C heteronuclear couplings were measured using a ^1^H-^13^C *J*-HMBC experiment, thus providing complimentary evidence for correct structural assignment. The *J*-HMBC spectrum showed a strong correlation between the methylene proton δ 4.78 ppm and the quaternary carbon C-5a with an 8.0-hertz coupling constant, while the proton δ 4.66 ppm correlated very weakly, with a *J* value of only 2.2 Hz, which was attributed to a ^5^*J*_C-5a, H-3_. Finally, the pyrazole 8-H proton (δ 8.13 ppm) not only exhibited long-range HMBC correlations with neighboring N-7 “pyrrole-like” (δ −177.4 ppm) and N-6 “pyridine-like” (δ −117.7 ppm) nitrogen atoms, but also with the C-5a, C-8a and C-8b quaternary carbons, which were unambiguously assigned with the subsequent 1,1-ADEQUATE experiment, thus allowing all the heterocyclic moieties to be connected together. The structure of compounds **5b**–**d** was determined by analogous NMR spectroscopy experiments, as described above. The skeleton of the pyrazolo[4′,3′:5,6]pyrano[4,3-*c*][1,2]oxazole ring system contains three nitrogen atoms. The chemical shifts of the N-1, N-6 and N-7 atoms of compounds **5a**–**c** were in a range from δ −30.9 to −32.2, δ −116.9 to −117.7 and δ −177.4 to −179.5 ppm, respectively, while in the case of compound **5d**, which lacked a phenyl moiety at site 7, the chemical shifts of N-1, N-6 and N-7 atoms were δ −35.8, δ −112.3 and δ −194.4 ppm, respectively.

In the case of compound **6**, a comparison of the ^1^H NMR spectra between **5a** and **6** clearly indicated the disappearance of methine 3a-H (δ 3.86–3.91 ppm) and methylene 3-H protons (δ 4.66 and 3.79 ppm) and the formation of a new downfield methine 3-H proton signal at δ 8.21 ppm. The aforementioned methine proton that appeared as a triplet was mutually coupled with methylene 4-H protons (doublet, δ 5.41 ppm), as indicated by their *meta*-coupling (^4^*J_HH_* = 1.3 Hz). Moreover, a comparison between the ^1^H-^1^H COSY and ^1^H-^1^H NOESY spectra showed a complete absence of COSY cross-peaks between 3-H and 4-H and only strong NOEs, which confirmed their proximity in space. This finding strongly hinted at a neighboring quaternary carbon at site 3a, which was unambiguously assigned from 1,1-ADEQUATE spectral data, where the protonated carbons C-3 (δ 150.7 ppm) and C-4 (δ 63.3 ppm) showed a sole correlation with C-3a at δ 109.8 ppm. As expected, the ^15^N chemical shifts of N-6 (δ −116.3) and N-7 (δ −179.6) atoms were highly comparable to those of compounds **5a**–**c**; only the N-1 atoms were slightly different and resonated at δ −20.4 ppm, which is in good agreement with the data reported in the literature [60].

To expand the structural diversity of the obtained 3a,4-dihydro-3*H*,7*H*-pyrazolo[4′,3′:5,6]pyrano[4,3-*c*][1,2]oxazole system, we prepared additional *vic*-cinnamyloxy-oxime **9** as a substrate for the INOC reaction (Figure 2). As the cinnamyloxy group turned out to be sensitive towards Vilsmeier–Haack reaction conditions, the *O*-alkylation formylation sequence of compound **1a** successfully applied to the synthesis of 3-allyloxypyrazole-4-carbaldehydes **3a**–**d** was reorganized. In short, first, the hydroxy group of pyrazol-3-ol (**1a**) was transformed to a benzyloxy group; then, the obtained 3-benzyloxypyrazole was formylated under the Vilsmeier–Haack reaction conditions, and the protecting OBn group was cleaved by TFA to give 3-hydroxy-1*H*-pyrazole-4-carbaldehyde **7** [35]. The latter compound was subjected to an alkylation reaction with cinnamyl chloride and the appropriate 3-cinnamyloxy-1*H*-pyrazole-4-carbaldehyde (**8**) was obtained in very good (82%) yield. A subsequent reaction of **8** with hydroxylamine gave the aldoxime **9**, which was successfully used for the INOC reaction, and 3-phenyl-3a,4-dihydro-3*H*,7*H*-pyrazolo[4′,3′:5,6]pyrano[4,3-*c*][1,2]oxazole *trans*-**10** was obtained with a fair (62%) yield.

While the structural elucidation of compound *trans*-**10** was straightforward and followed the same logical approach as in the case of compounds **5a**–**d** and **6**, determination of the relative configuration at C-3 and C-3a proved to be a more challenging task and was achieved by combined analysis of NOESY, *J*-coupling and molecular modeling data. For instance, the initial geometry optimizations were performed using MM2 and MMFF94 force fields [61], followed by DFT methods using B3LYP/def2-TZVP, as implemented in ORCA 5.0.0 [62], which provided the dihedral angle values between H-C(3)-C(3a)-H for structures *trans*-**10** (154.34°) and *cis*-**10** (19.28°). Then, the theoretical ^1^H-^1^H coupling constants were calculated with the same software package following a standard procedure using a B3LYP/PCSSEG-2 basis set. The dihedral angle values were used in the calculation of ^3^*J_H_*_3,*H*3a_ by the Haasnoot–de Leeuw–Altona (HLA) equation [63]. The ^3^*J_H_*_3,*H*3a_ values estimated by the HLA method were 10.0 Hz for *trans*-**10** and 8.2 Hz for *cis*-**10**, while ORCA 5.0.0 calculations were 13.4 and 10.8 Hz, respectively. The experimental value 13.1 Hz, which was obtained from the ^1^H NMR spectrum, hinted in favor of the *trans*-**10** structure. A highly similar class of heterocycles, naphthopyranoisoxazolines, were synthesized by Liaskopoulos et al. [64], where the target compounds possessed a *trans* configuration, as confirmed by X-ray and NMR analyses, and their ^3^*J_H_*_3,*H*3a_ values were in the range of 12.2–12.5 Hz. Finally, unambiguous confirmation of *trans*-**10** assignment was obtained from the ^1^H-^1^H NOESY spectrum, as it was evident from the geometrically optimized structures (Appendix A) that, in the case of *cis*-**10**, there should be a strong NOE between protons 3-H and 3a-H, while the NOE between 3a-H and the neighboring 3-phenyl group aromatic protons is not possible. However, in our case, the ^1^H-^1^H NOESY spectrum showed completely opposite measurements. Moreover, a distinct NOE between protons 3-H/4-H_a_ and 3a-H/4-H_b_ is only possible if the relative configuration is *trans*-**10**.

We also investigated the INOC reaction of *vic*–alkyne–oxime substrates **12** and **14a**–**c** (Figure 3). To obtain the intermediate compound **12**, firstly, 3-hydroxypyrazole **1a** was *O*-propargylated and formylated to give carbaldehyde **11** [26]. Compound **11** was then successfully converted to 4*H*,7*H*-pyrazolo[4′,3′:5,6]pyrano[4,3-*c*][1,2]oxazole **6** via the INOC reaction of intermediate oxime **12**, and the targeted new polyheterocyclic compound **6** was obtained in good (79%) yield. In addition, alkyne **11** was further subjected to the Sonogashira cross-coupling reaction with various (het)arylhalides, i.e., iodobenzene, 1-iodonaphthalene and 2-bromopyridine, under the standard Sonogashira cross-coupling reaction conditions (Pd(PPh_3_)_2_Cl_2_, CuI, DMF, 60 °C, argon atmosphere) to give alkynes **13a**–**c** in good yields [26]. Compounds **13a**–**c** were further treated with hydroxylamine hydrochloride to provide aldoximes **14a**–**c**, which were used in the INOC reaction without further purification. Aldoxime **14a** was subjected to a detailed NMR analysis, and, to our delight, it was obtained as a pure *syn* isomer, which was easily elucidated from a ^1^*J*_CH_ coupling constant of the iminyl moiety, which was 179.2 Hz. Moreover, 4*H*,7*H*-pyrazolo[4′,3′:5,6]pyrano[4,3-*c*][1,2]oxazoles **15a**–**c** were obtained in good yields.

As expected, the chemical shifts of the 3-aryl-substituted compounds **15a**–**c** were highly similar to those of compound **6**. A distinct difference in the ^1^H NMR spectra of the aforementioned compounds was that they contained only a singlet for the methylene 4-H protons in the area of δ 5.32–6.03 ppm, which indicated the lack of coupling partners. The data from the ^1^H-^13^C HMBC spectra revealed a distinct long-range correlation between the aforementioned methylene protons and a quaternary carbon at site 3. Moreover, the protons from a neighboring 3-aryl moiety shared an HMBC cross-peak with carbon C-3 as well, thus allowing different structural fragments to be joined together. The chemical shifts of the N-1, N-6 and N-7 atoms of 3-aryl-substituted compounds were in ranges of δ −23.9 to −25.0, δ −116.4 to −117.4 and δ −179.6 to −180.1 ppm, respectively, while, in the case of compound **15c** with a pyridin-2-yl moiety, the pyridine nitrogen resonated at δ −72.8 ppm.

## 3. Materials and Methods

### 3.1. General Information

All starting materials were purchased from commercial suppliers and were used as received. Flash column chromatography was performed on Silica Gel 60 Å (230–400 µm, Merck). Thin-layer chromatography was carried out on Silica Gel plates (Merck Kieselgel 60 F_254_) and visualized by UV light (254 nm). Melting points were determined on a Büchi M-565 melting point apparatus and were uncorrected. The IR spectra were recorded on a Bruker Vertex 70v FT-IR spectrometer using neat samples and are reported in frequency of absorption (cm^−1^). Mass spectra were obtained on a Shimadzu LCMS-2020 (ESI^+^) spectrometer. High-resolution mass spectra were measured on a Bruker MicrOTOF-Q III (ESI^+^) apparatus. The ^1^H, ^13^C and ^15^N NMR spectra were recorded in CDCl_3_, DMSO-*d*_6_ or TFA-*d* solutions at 25 °C on a Bruker Avance III 700 (700 MHz for ^1^H, 176 MHz for ^13^C, 71 MHz for ^15^N) spectrometer equipped with a 5 mm TCI ^1^H-^13^C/^15^N/D z-gradient cryoprobe. The chemical shifts (δ), expressed in ppm, were relative to tetramethylsilane (TMS). The ^15^N NMR spectra were referenced to neat, external nitromethane (coaxial capillary). Full and unambiguous assignment of the ^1^H, ^13^C and ^15^N NMR resonances was achieved using a combination of standard NMR spectroscopic techniques [65] such as DEPT, COSY, TOCSY, NOESY, gs-HSQC, gs-HMBC, H2BC, LR-HSQMBC and 1,1-ADEQUATE experiments [66]. Structures for molecular modeling were built using Chem3D Pro 17.0, and were optimized by MM2 and MMFF94 force fields, followed by DFT methods using B3LYP/def2-TZVP for dihedral angle measurements, and B3LYP/PCSSEG-2 for the calculation of theoretical ^1^H-^1^H coupling constants, using a standard procedure as implemented in the ORCA 5.0.0 software package. The dihedral angle values were used in the calculation of vicinal ^1^H-^1^H coupling constants using the Mestre-J 1.1 software [67] and HLA (general, beta effect) equation. ^1^H-, ^13^C-, and ^1^H-^15^N HMBC NMR spectra, and HRMS data of new compounds, are provided in Appendix A.

### 3.2. Synthesis of 1-Methyl-3-[(prop-2-en-1-yl)oxy]-1H-pyrazole *(**2d**)*

A solution of 3-hydroxypyrazole **1d** (450 mg, 2.5 mmol) in dry DMF (5 mL) was cooled to 0 °C under an inert atmosphere, and NaH (60% dispersion in mineral oil, 60 mg, 2.5 mmol) was added portion-wise. After stirring the reaction mixture for 15 min, allyl bromide (370 mg, 3 mmol) was added dropwise. The mixture was stirred at 60 °C for 1 h, then poured into water and extracted with ethyl acetate (3 × 10 mL). The organic layers were combined, washed with brine, dried over Na_2_SO_4_, filtrated, and the solvent was evaporated. The residue was purified by flash column chromatography (SiO_2_, eluent: ethyl acetate/*n*-hexane, 1:7, *v*/*v*) to provide the desired compound **2d** as a brown liquid, yield 330 mg, 96%. IR (v_max_, cm^−1^): 2931 (CH_aliph_), 1728, 1694, 1538, 1494, 1410, 1346, 1224 (–C=C, C=C, C–N, C–O–C). ^1^H NMR (700 MHz, CDCl_3_): δ 3.73 (s, 3H, CH_3_), 4.65 (dt, *J* = 5.5, 1.5 Hz, 2H, OCH_2_CHCH_2_), 5.24 (dq, *J* = 10.6, 1.6 Hz, 1H, OCH_2_CHCH_2_), 5.40 (dq, *J* = 17.2, 1.6 Hz, 1H, OCH_2_CHCH_2_), 5.62 (d, *J* = 2.3 Hz, 1H, 4-H), 6.06 (ddt, *J* = 17.2, 10.8, 5.5 Hz, 1H, OCH_2_CHCH_2_), 7.12 (d, *J* = 2.3 Hz, 1H, 5-H). ^13^C NMR (176 MHz, CDCl_3_): δ 38.9 (CH_3_), 69.7 (OCH_2_CHCH_2_), 90.2 (C-4), 117.4 (OCH_2_CHCH_2_), 131.2 (C-5), 133.4 (OCH_2_CHCH_2_), 163.0 (C-3). MS *m*/*z* (%): 139 ([M + H]^+^, 100). HRMS (ESI^+^) for C_7_H_10_N_2_NaO ([M + Na]^+^) requires 161.0685, found 161.0685.

### 3.3. Synthesis of 1-Methyl-3-[(prop-2-en-1-yl)oxy]-1H-pyrazole-4-carbaldehyde *(**3d**)*

Phosphorus oxychloride (0.2 mL, 2.5 mmol) was added dropwise to DMF (0.23 mL, 2.5 mmol) at −10 °C temperature. Then, pyrazole **2d** (0.62 mmol) was added to the Vilsmeier—Haack complex, and the reaction mixture was heated at 70 °C temperature for 1 h. After neutralization with 10% NaHCO_3_ solution, it was extracted with ethyl acetate (3 × 10 mL). The organic layers were combined, washed with brine, dried over Na_2_SO_4_, filtrated, and the solvent was evaporated. The residue was purified by flash column chromatography (SiO_2_, eluent: ethyl acetate/*n*-hexane, 1:6, *v*/*v*) to provide the desired compound **3c** as a brown solid, yield 400 mg, 95%, mp 59–60 °C. IR (KBr, v_max_, cm^−1^): 3122, 3088 (CH_arom_), 2943, 2823 (CH_aliph_), 1673 (C=O), 1568, 1507, 1492, 1429, 1404, 1343, 1202, 1179, 1009 (C=N, C=C, C–N, C–O–C). ^1^H NMR (700 MHz, CDCl_3_): δ 3.78 (s, 3H, CH_3_), 4.79 (dt, *J* = 5.6, 1.5 Hz, 2H, OCH_2_), 5.29 (dq, *J* = 10.5, 1.4 Hz, 1H, OCH_2_CHCH_2_), 5.43 (dq, *J* = 17.2, 1.6 Hz, 1H, OCH_2_CHCH_2_), 6.09 (ddt, *J* = 17.2, 10.5, 5.6 Hz, 1H, OCH_2_CHCH_2_), 7.69 (s, 1H, 5-H), 9.75 (s, 1H, CHO). ^13^C NMR (176 MHz, CDCl_3_): δ 39.7 (CH_3_), 69.9 (OCH_2_CHCH_2_), 109.5 (C-4), 118.3 (OCH_2_CHCH_2_), 132.6 (C-5), 133.4 (OCH_2_CHCH_2_), 163.1 (C-3), 183.0 (CHO). MS *m*/*z* (%): 167 ([M + H]^+^, 100). HRMS (ESI^+^) for C_8_H_10_N_2_NaO_2_ ([M + Na]^+^) requires 189.0634, found 189.0634

### 3.4. General Procedure for the Oxime Formation Reaction from 1H-Pyrazole-4-carbaldehydes ***3a**–**d***

To a solution of appropriate 3-[(prop-2-en-1-yl)oxy]-1*H*-pyrazole-4-carbaldehyde **3a**–**d** (3 mmol) in EtOH (10 mL), sodium acetate (369 mg, 4.5 mmol) and hydroxylamine hydrochloride (250 mg, 3.6 mmol) were added, and the reaction mixture was refluxed for 15 min. After completion of the reaction as monitored by TLC, EtOH was evaporated, and the mixture was diluted with water (10 mL) and extracted with ethyl acetate (3 × 10 mL). The organic layers were combined, washed with brine, dried over Na_2_SO_4_, filtrated, and the solvent was evaporated. The residue was purified by flash column chromatography (SiO_2_, eluent: ethyl acetate/*n*-hexane, 1:6, *v*/*v*) to provide the desired compounds **4a**–**c**, as mixtures of *syn*-*anti* isomers or pure *syn* isomer **4d**. Due to small extent of minor isomer and heavy overlap with signals of the predominant *syn-(Z)* isomer, NMR spectroscopy data of the major isomer only are presented, while the relevant NMR spectroscopy data of the minor isomer are presented in a Appendix A.

#### 3.4.1. N-[(Z/E)-{1-Phenyl-3-[(prop-2-en-1-yl)oxy]-1H-pyrazol-4-yl}methylidene] Hydroxylamine (**4a**)

**4a** was obtained as a mixture of *syn*- and *anti*- isomers in ratio *syn*-**4a**:*anti*-**4a** 97:3. Yellow solid, yield 707 mg, 97%, mp 103–104 °C. IR (v_max_, cm^−1^): 3272 (OH), 3164, 3126, 3067, 3028 (CH_arom_), 2967, 2935 (CH_aliph_), 1641 (C=N), 1557, 1466, 1344, 1220 (C=C, C–N, C–O–C), 647, 754 (CH=CH of benzene). ^1^H NMR (700 MHz, DMSO-*d*_6_): δ 4.86 (dt, *J* = 5.4, 1.4 Hz, 2H, OCH_2_), 5.31 (dq, *J* = 10.5, 1.4 Hz, 1H, OCH_2_CHCH_2_), 5.48 (dq, *J* = 17.2, 1.6 Hz, 1H, OCH_2_CHCH_2_), 6.14 (ddt, *J* = 17.2, 10.7, 5.4 Hz, 1H, OCH_2_CHCH_2_), 7.26–7.31 (m, 2H, Ph 4-H, CHNOH), 7.44–7.51 (m, 2H, Ph 3,5-H), 7.77–7.84 (m, 2H, Ph 2,6-H), 8.85 (s, 1H, Pz 5-H), 11.63 (s, 1H, OH). ^13^C NMR (176 MHz, DMSO-*d*_6_): 69.3 (OCH_2_), 100.5 (Pz C-4), 117.8 (Ph C-2,6), 118.0 (OCH_2_CHCH_2_), 125.9 (Ph C-4), 129.5 (Ph C-3,5), 131.0 (Pz C-5), 133.1 (OCH_2_CHCH_2_), 134.9 (CH=N–OH), 139.1 (Ph C-1), 161.3 (Pz C-3). ^15^N NMR (71 MHz, DMSO-*d*_6_): δ −184.1 (N-1), −123.0 (N-2), −19.2 (CH=N–OH). MS *m*/*z* (%): 244 ([M + H]^+^, 100). HRMS (ESI^+^) for C_13_H_14_N_3_O_2_ ([M + H]^+^) requires 244.1081, found 244.1081.

#### 3.4.2. N-[(Z/E)-[4-Fluorophenyl)-3-[(prop-2-en-1-yl)oxy}-1H-pyrazol-4-yl}methylidene] Hydroxylamine (**4b**)

**4b** was obtained as a mixture of *syn*- and *anti*- isomers in ratio *syn*-**4b**:*anti*-**4b** 91:9. Yellow solid, yield 689 mg, 88%, mp 134–136 °C. IR (v_max_, cm^−1^): 3229 (OH), 3175, 3160, 3097 (CH_arom_), 2924 (CH_aliph_), 1669, 1564, 1502, 1390, 1224, 1209 (C=N, C=C, C–N, C–O–C, C–F), 941 (CH=CH of benzene). ^1^H NMR (700 MHz, DMSO-*d*_6_): δ 4.86 (dt, *J* = 5.4, 1.4 Hz, 2H, OCH_2_), 5.31 (dq, *J* = 10.5, 1.3 Hz, 1H, OCH_2_CHCH_2_), 5.46 (dq, *J* = 17.3, 1.6 Hz, 1H, OCH_2_CHCH_2_), 6.14 (ddt, *J* = 17.3, 10.7, 5.4 Hz, 1H, OCH_2_CHCH_2_), 7.28 (s, 1H, CHNOH), 7.30–7.34 (m, 2H, Ph 3,5-H), 7.83–7.86 (m, 2H, Ph 2,6-H), 8.84 (s, 1H, Pz 5-H), 11.64 (s, 1H, OH). ^13^C NMR (176 MHz, DMSO-*d*_6_): δ 69.3 (OCH_2_), 100.5 (Pz C-4), 116.17 (d, ^2^*J_C_*_,*F*_ = 22.9 Hz, Ph C-3,5), 118.0 (OCH_2_CHCH_2_), 119.89 (d, ^3^*J_C_*_,*F*_ = 8.3 Hz, Ph C-2,6), 131.2 (Pz C-5), 133.1 (OCH_2_CHCH_2_), 134.9 (CH=N–OH), 135.73 (d, ^4^*J_C_*_,*F*_ = 2.5 Hz, Ph C-1), 160.03 (d, ^1^*J_C_*_,*F*_ = 242.8 Hz, Ph C-4), 161.3 (Pz C-3). ^15^N NMR (71 MHz, DMSO-*d*_6_): δ −185.8 (N-1), −122.6 (N-2), −19.2 (CH=N–OH). MS *m*/*z* (%): 262 ([M + H]^+^, 100). HRMS (ESI^+^) for C_13_H_13_FN_3_O_2_ ([M + H]^+^) requires 262.0986, found 262.0986.

#### 3.4.3. N-[(Z/E)-[4-Bromophenyl)-3-[(prop-2-en-1-yl)oxy}-1H-pyrazol-4-yl}methylidene] Hydroxylamine (**4c**)

**4c** was obtained as a mixture of *syn*- and *anti*- isomers in ratio *syn*-**4c**:*anti*-**4c** 99:1. Yellowish solid, yield 840 mg, 87%, mp 125.5–127 °C. IR (KBr, v_max_, cm^−1^): 3173 (OH), 3076, 3033, 3021 (CH_arom_), 2848 (CH_aliph_), 1656, 1589, 1508, 1508, 1452, 1424, 1404, 1395, 1348, 1220, 1187, 1008 (C=N, C=C, C–N, C–O–C), 995, 955, 934, 890, 819, 803, 716 (C–Br, CH=CH of benzene). ^1^H NMR (700 MHz, DMSO-*d*_6_): δ 4.85 (dt, *J* = 5.5, 1.5 Hz, 2H, OCH_2_), 5.30 (dq, *J* = 10.5, 1.5 Hz, 1H, OCH_2_CHCH_2_), 5.48 (dq, *J* = 17.3, 1.6 Hz, 1H, OCH_2_CHCH_2_), 6.13 (ddt, *J* = 17.1, 10.7, 5.4 Hz, 1H, OCH_2_CHCH_2_), 7.27 (s, 1H, CHNOH), 7.62–7.67 (m, 2H, Ph 2,6-H), 7.76–7.80 (m, 2H, Ph 3,5-H), 8.88 (s, 1H, Pz 5-H), 11.67 (s, 1H, OH). ^13^C NMR (176 MHz, DMSO-*d*_6_): δ 69.4 (OCH_2_), 100.9 (Pz C-4), 118.05 (Ph C-4), 118.09 (OCH_2_CHCH_2_), 119.7 (Ph C-2,6), 131.2 (Pz C-5), 132.3 (Ph C-3,5), 133.1 (OCH_2_CHCH_2_), 134.7 (CH=N-OH), 138.4 (Ph C-4), 161.4 (Pz C-3). ^15^N NMR (71 MHz, DMSO-*d*_6_): δ −186.1 (N-1), −123.6 (N-2), −18.2 (CH=N–OH). MS *m*/*z* (%): 322 ([M + H]^+^, 100); 324 ([M + H + 2]^+^, 100). HRMS (ESI^+^) for C_13_H_12_BrN_3_NaO_2_ ([M + Na]^+^) requires 344.0005, found 344.0008.

#### 3.4.4. N-[(E)-{1-Methyl-3-[(prop-2-en-1-yl)oxy]-1H-pyrazol-4-yl}methylidene] Hydroxylamine (**4d**)

Yellow solid, yield 448 mg, 82%, mp 88–89 °C. IR (v_max_, cm^−1^): 3145 (OH), 3089 (CH_arom_), 2992, 2849, 2821 (CH_aliph_), 1648, 1559, 1490, 1417, 1342, 1177 (C=N, C=C, C–N, C–O–C). ^1^H NMR (700 MHz, DMSO-*d*_6_): δ 3.71 (s, 3H, CH_3_), 4.68 (dt, *J* = 5.3, 1.4 Hz, 2H, OCH_2_), 5.25 (dq, *J* = 10.5, 1.5 Hz, 1H, OCH_2_CHCH_2_), 5.39 (dq, *J* = 17.3, 1.7 Hz, 1H, OCH_2_CHCH_2_), 6.05 (ddt, *J* = 17.2, 10.6, 5.3 Hz, 1H, OCH_2_CHCH_2_), 7.13 (s, 1H, CHNOH), 8.19 (s, 1H, Pz 5-H), 11.19 (s, 1H, OH). ^13^C NMR (176 MHz, DMSO-*d*_6_): δ 38.6 (CH_3_), 69.0 (OCH_2_), 97.3 (Pz C-4), 117.5 (OCH_2_CHCH_2_), 133.5 (OCH_2_CHCH_2_), 135.2 (Pz C-5), 135.6 (CH=N–OH), 160.1 (Pz C-3). ^15^N NMR (71 MHz, DMSO-*d*_6_): δ −199.0 (N-1), −114.8 (N-2), −25.7 (CH=N–OH). MS *m*/*z* (%): 182 ([M + H]^+^, 100). HRMS (ESI^+^) for C_8_H_12_N_3_O_2_ ([M + H]^+^) requires 182.0924, found 182.0924. C_8_H_11_N_3_NaO_2_ ([M + Na]^+^) requires 204.0743, found 204.0743.

### 3.5. General Procedure for the Cycloaddition Reaction of Pyrazole Oximes ***4a**–**d***

Into the solution of appropriate pyrazole (0.4 mmol) **4a**–**d** in DCM (5 mL), sodium hypochlorite (10% aq. solution, 0.5 mL, 0.8 mmol) was added, and the reaction mixture was stirred for 1 h at room temperature. After completion of the reaction as monitored by TLC, it was diluted with water (10 mL) and extracted with DCM (3 × 10 mL). The organic layers were combined, washed with brine, dried over Na_2_SO_4_, filtrated, and the solvent was evaporated. The residue was purified by flash column chromatography (SiO_2_, eluent: ethyl acetate/*n*-hexane, 1:4, *v*/*v*) to provide the desired compounds **5a**–**d**.

#### 3.5.1. 7-Phenyl-3a,4-dihydro-3H,7H-pyrazolo[4′,3′:5,6]pyrano[4,3-c][1,2]oxazole (**5a**)

White solid, yield 66 mg, 68%, mp 179–180 °C. IR (v_max_, cm^−1^): 3129 (CH_arom_), 2953, 2922, 2852 (CH_aliph_), 1576, 1503, 1485, 1411, 1360, 1262, 1209, 1051 1177 (C=N, C=C, C–N, C–O–C, N–O), 813, 685 (CH=CH of benzene). ^1^H NMR (700 MHz, CDCl_3_): δ 3.79 (dd, *J* = 13.8, 8.0 Hz, 1H, 3-H), 3.86–3.91 (m, 1H, 3a-H), 4.17 (dd, *J* = 12.1, 10.6 Hz, 1H, 4-H), 4.66 (dd, *J* = 9.4, 8.1 Hz, 1H, 3-H), 4.78 (dd, *J* = 10.5, 5.5 Hz, 1H, 4-H), 7.29 (t, *J* = 7.4 Hz, 1H, Ph 4-H), 7.43 (t, *J* = 8.0 Hz, 2H, Ph 3,5-H), 7.63 (d, *J* = 7.7 Hz, 2H, Ph 2,6-H), 8.13 (s, 1H, 8-H). ^13^C NMR (176 MHz, CDCl_3_): δ 46.7 (C-3a), 69.7 (C-3), 70.9 (C-4), 96.5 (C-8a), 118.9 (Ph C-2,6), 123.6 (C-8), 127.1 (Ph C-4), 129.6 (Ph C-3,5), 139.4 (Ph C-1), 149.4 (C-8b), 162.8 (C-5a). ^15^N NMR (71 MHz, CDCl_3_): δ −177.4 (N-7), −117.7 (N-6), −32.2 (N-1). MS *m*/*z* (%): 242 ([M + H]^+^, 100). HRMS (ESI^+^) for C_13_H_11_N_3_NaO_2_ ([M + Na]^+^) requires 264.0743, found 264.0744.

#### 3.5.2. 7-(4-Fluorophenyl)-3a,4-dihydro-3H,7H-pyrazolo[4′,3′:5,6]pyrano[4,3-c][1,2]oxazole (**5b**)

White solid, yield 65 mg, 63%, mp 214–215 °C. IR (v_max_,cm^−1^): 3102 (CH_arom_), 1644, 1582, 1513, 1489, 1363, 1218 (C=N, C=C, C–N, C–O–C, N–O, C–F), 832, 600 (CH=CH of benzene). ^1^H NMR (700 MHz, CDCl_3_): δ 3.81 (dd, *J* = 13.8, 8.0 Hz, 1H, 3-H), 3.87–3.93 (m, 1H, 3a-H), 4.18 (dd, *J* = 12.0, 10.7 Hz, 1H, 4-H), 4.68 (dd, *J* = 10.5, 5.5 Hz, 1H, 3-H), 4.80 (dd, *J* = 10.5, 5.5 Hz, 1H, 4-H), 7.13–7.16 (m, 2H, Ph 3,5-H), 7.60–7.62 (m, 2H, Ph 2,6-H), 8.07 (s, 1H, 8-H). ^13^C NMR (176 MHz, CDCl_3_): δ 46.7 (C-3a), 69.7 (C-3), 71.0 (C-4), 96.6 (C-8a), 116.6 (d, ^2^*J_C_*_,*F*_ = 23.3 Hz, Ph C-3,5), 120.8 (d, ^3^*J_C_*_,*F*_ = 8.0 Hz, Ph C-2,6), 123.7 (C-8), 135.8 (d, ^4^*J_C_*_,*F*_ = 2.4 Hz, Ph C-1), 149.9 (C-8b), 161.5 (d, ^1^*J_C_*_,*F*_ = 247.1 Hz, Ph C-4), 162.9 (C-5a). ^15^N NMR (71 MHz, CDCl_3_): δ −179.2 (N-7), −116.9 (N-6), −31.9 (N-1). MS *m*/*z* (%): 260 ([M + H]^+^, 100). HRMS (ESI^+^) for C_13_H_10_FN_3_NaO_2_ ([M + Na]^+^) requires 282.0649, found 282.0648.

#### 3.5.3. 7-(4-Bromophenyl)-3a,4-dihydro-3H,7H-pyrazolo[4′,3′:5,6]pyrano[4,3-c][1,2]oxazole (**5c**) 

White solid, yield 82 mg, 64%, mp 255–255.5 °C. IR (KBr, v_max_, cm^−1^): 3129 (CH_arom_), 1648, 1590, 1575, 1487, 1452, 1419, 1364, 1263, 1207, 1073, 1051, 1007 (C=N, C=C, C–N, C–O–C), 982, 942, 832, 713, 498 (C–Br, CH=CH of benzene). ^1^H NMR (700 MHz, CDCl_3_): δ 3.81 (dd, *J* = 13.8, 8.0 Hz, 1H, 3-H), 3.87–3.93 (m, 1H, 3a-H), 4.18 (dd, *J* = 12.1, 10.6 Hz, 1H, 4-H), 4.68 (dd, *J* = 9.4, 8.1 Hz, 1H, 3-H), 4.80 (dd, *J* = 10.6, 5.5 Hz, 1H, 4-H), 7.52–7.54 (m, 2H, Ph 2,6-H), 7.57–7.58 (m, 2H, Ph 3,5-H), 8.12 (s, 1H, 8-H). ^13^C NMR (176 MHz, CDCl_3_): δ 46.7 (C-3a), 69.8 (C-3), 71.0 (C-4), 97.0 (C-8a), 120.3 (Ph C-2,6), 120.4 (C-8), 123.5 (C-8), 132.8 (Ph C-3,5), 138.4 (Ph C-1), 149.2 (C-8b), 162.9 (C-5a). ^15^N NMR (71 MHz, CDCl_3_): δ −179.5 (N-7), −117.6 (N-6), −30.9 (N-1). MS *m*/*z* (%): 320 ([M]^+^, 100); 322 ([M + 2]^+^, 100). HRMS (ESI^+^) for C_13_H_10_BrN_3_NaO_2_ ([M + Na]^+^) requires 341.9849, found 341.9848.

#### 3.5.4. 7-Methyl-3a,4-dihydro-3H,7H-pyrazolo[4′,3′:5,6]pyrano[4,3-c][1,2]oxazole (**5d**)

White solid, yield 30 mg, 42%, mp 159–160 °C. IR (v_max_, cm^−1^): 3144, 3110 (CH_arom_), 2996, 2946, 2867 (CH_aliph_), 1645, 1570, 1490, 1457, 1363, 1246, 1169, 1079, 1016 (C=N, C=C, C–N, C–O–C, N–O). ^1^H NMR (700 MHz, CDCl_3_): δ 3.67 (dd, *J* = 13.8, 8.1 Hz, 1H, 3-H), 3.75 (s, 3H, CH_3_), 3.76–3.80 (m, 1H, 3a-H), 4.03 (dd, *J* = 12.1, 10.6 Hz, 1H, 4-H), 4.55 (dd, *J* = 9.4, 8.1 Hz, 1H, 3-H), 4.65 (dd, *J* = 10.5, 5.4 Hz, 1H, 4-H), 7.49 (s, 1H, 8-H). ^13^C NMR (176 MHz, CDCl_3_): δ 39.7 (CH_3_), 47.0 (C-3a), 69.5 (C-3), 70.8 (C-4), 93.8 (C-8a), 127.4 (C-8), 149.8 (C-8b), 162.0 (C-5a). ^15^N NMR (71 MHz, CDCl_3_): δ −194.4 (N-7), −112.3 (N-6), −35.8 (N-1). MS *m*/*z* (%): 180 ([M + H]^+^, 100). HRMS (ESI^+^) for C_8_H_9_N_3_NaO_2_ ([M + Na]^+^) requires 202.0587, found 202.0586.

### 3.6. Oxidation of 7-Phenyl-3,3a,4,7-tetrahydropyrazolo[4′,3′:5,6]pyrano[4,3-c]oxazole (***5a***) with MnO_2_ to 7-Phenyl-4,7-dihydropyrazolo[4′,3′:5,6]pyrano[4,3-c]oxazole *(**6**)*

To a flask adapted with a Dean–Stark collector, 7-phenyl-3,3a,4,7-tetrahydropyrazolo[4′,3′:5,6]pyrano[4,3-*c*]oxazole (116 mg, 0.48 mmol), toluene (5 mL) and MnO_2_ (754 mg, 6 mmol) were added. The mixture was heated under reflux for 4 h, cooled to room temperature, filtered over celite, and concentrated under reduced pressure. The residue was purified by flash column chromatography (SiO_2_, eluent: dichloromethane/methanol, 100:1, *v*/*v*) to provide the desired compound as a colorless crystal, yield 43 mg, 38%, mp 205–208 °C. IR (KBr, ν_max_, cm^−1^): 3100 (CH_arom_), 1642, 1594, 1521, 1468, 1396, 1252, 1040 (C=N, C=C, C–N, C–O–C, N–O), 944, 801, 769, 684 (CH=CH of benzene). ^1^H NMR (700 MHz, CDCl_3_): δ 5.41 (d, *J* = 1.3 Hz, 2H, CH_2_), 7.29–7.32 (m, 1H, Ph 4−H), 7.44–7.48 (m, 2H, Ph 3,5-H), 7.66–7.69 (m, 2H, Ph 2,6-H), 8.20 (s, 1H, 8-H), 8.21 (t, *J* = 1.3 Hz, 1H, 3-H). ^13^C NMR (176 MHz, CDCl_3_): δ 63.3 (CH_2_), 96.5 (C-8a), 109.8 (C-3a), 118.6 (Ph C-2,6), 122.6 (C-8), 126.8 (Ph C-4), 129.6 (Ph C-3,5), 139.4 (Ph C-1), 150.7 (C-3), 151.0 (C-8b), 162.4 (C-5a). ^15^N NMR (71 MHz, CDCl_3_): δ −179.6 (N-7), −116.3 (N-6), −20.4 (N-1). MS *m*/*z* (%): 240 ([M + H]^+^, 100). HRMS (ESI^+^) for C_13_H_9_N_3_NaO_2_ ([M + Na]^+^) requires 262.0587, found 262.0589.

### 3.7. Synthesis of 1-Phenyl-3-{[(2Z)-3-phenylprop-2-en-1-yl]oxy}-1H-pyrazole-4-carbaldehyde *(**8**)*

A solution of 3-hydroxy-1-phenyl-1*H*-pyrazole-4-carbaldehyde **7** (188 mg, 1 mmol) in dry DMF (2 mL) was cooled to 0 °C under an inert atmosphere, and NaH (60% dispersion in mineral oil, 40 mg, 1 mmol) was added portion-wise. After stirring the reaction mixture for 15 min, cinnamyl chloride (185 mg, 1.2 mmol) was added dropwise. The mixture was stirred at 70 °C for 15 min, then poured into water and extracted with ethyl acetate (3 × 10 mL). The organic layers were combined, washed with brine, dried over Na_2_SO_4_, filtrated, and the solvent was evaporated. The residue was purified by flash column chromatography (SiO_2_, eluent: ethyl acetate/*n*-hexane, 1:4, *v*/*v*) to provide the desired compound as a brown solid, yield 250 mg, 82%, mp 118–119 °C. IR (v_max_, cm^−1^): 3096 (CH_arom_), 2821 (CH_aliph_), 1666 (C=O), 1555, 1499, 1357 (C=C, C–N, C–O–C), 964, 753, 686 (CH=CH of benzenes). ^1^H NMR (700 MHz, CDCl_3_): δ 5.00 (d, *J* = 6.3 Hz, 2H, OCH_2_CH=CH), 6.44 (dt, *J* = 15.9, 6.3 Hz, 1H, OCH_2_CH=CH), 6.72 (d, *J* = 15.9 Hz, 1H, OCH_2_CH=CH), 7.19 (t, *J* = 7.3 Hz, 1H, CPh 4-H), 7.23–7.27 (m, 3H, CPh 3,5-H and NPh 4-H), 7.36 (d, *J* = 7.6 Hz, 2H, CPh 2,6-H), 7.39 (t, *J* = 7.9 Hz, 2H, NPh 3,5-H), 7.58 (d, *J* = 8.0 Hz, 2H, NPh 2,6-H), 8.18 (s, 1H, 5-H), 9.82 (s, 1H, CHO). ^13^C NMR (176 MHz, CDCl_3_): δ 70.1 (OCH_2_CH=CH), 111.6 (C-4), 118.9 (NPh C-2,6), 123.6 (OCH_2_CH=CH), 126.8 (CPh C-2,6), 127.4 (NPh C-4), 128.2 (CPh C-4), 128.7 (CPh C-3,5), 129.58 (C-5), 129.64 (NPh C-3,5), 134.6 (OCH_2_CH=CH), 136.4 (CPh C-1), 139.1 (NPh C-1), 163.5 (C-3), 183.5 (CHO). MS *m*/*z* (%): 305 ([M + H]^+^, 100). HRMS (ESI^+^) for C_19_H_16_N_2_NaO_2_ ([M + Na]^+^) requires 327.1104, found 327.1104.

### 3.8. Synthesis of N-[(Z/E)-{1-Phenyl-3-{[(2Z)-3-phenylprop-2-en-1-yl)oxy]-1H-pyrazol-4-yl}methylidene]hydroxylamine* (**9**)*

This compound was synthesized in analogy to **4a**–**d**, except that pyrazole **8** was used as the adduct. Yellow solid, yield 823 mg, 86%, mp 129–130 °C. IR (v_max_, cm^−1^): 3161 (OH), 3022 (CH_arom_), 2922 (CH_aliph_), 1638, 1560, 1493, 1215 (C=N, C=C, C–N, C–O–C), 965, 745, 683 (CH=CH of benzene). ^1^H NMR (700 MHz, DMSO-*d*_6_): δ 5.03 (d, *J* = 5.4 Hz, 2H, OCH_2_), 6.60 (dt, *J* = 16.0, 6.1 Hz, 1H, OCH_2_CH=CH), 6.86 (d, *J* = 16.0 Hz, 1H, OCH_2_CH=CH), 7.26–7.30 (m, 2H, NPh 4-H, CPh 4-H), 7.32 (s, 1H, CHNOH), 7.36 (t, *J* = 7.6 Hz, 2H, CPh 3,5-H), 7.49 (t, *J* = 8.0 Hz, 2H, NPh 3,5-H), 7.52 (d, *J* = 7.4 Hz, 2H, CPh 2,6-H), 7.83 (d, *J* = 7.8 Hz, 2H, NPh 2,6-H), 8.86 (s, 1H, Pz 5-H), 11.63 (s, 1H, OH). ^13^C NMR (176 MHz, DMSO-*d*_6_): δ 69.3 (OCH_2_), 100.6 (Pz C-4), 117.8 (NPh C-2,6), 124.2 (OCH_2_CH=CH), 125.9 (NPh C-4), 126.6 (CPh C-2,6), 128.0 (CPh C-4), 128.7 (CPh C-3,5), 129.5 (NPh C-3,5), 131.0 (Pz C-5), 133.2 (OCH_2_CH=CH), 135.0 (CH=N–OH), 136.1 (CPh C-1), 139.1 (NPh C-1), 161.3 (Pz C-3). ^15^N NMR (71 MHz, DMSO-*d*_6_): δ −184.0 (N-1), −122.7 (N-2), −19.2 (CH=N–OH). MS *m*/*z* (%): 320 ([M + H]^+^, 100). HRMS (ESI^+^) for C_19_H_17_N_3_NaO_2_ ([M + Na]^+^) requires 342.1213, found 342.1213.

### 3.9. Synthesis of 3,7-Diphenyl-3a,4-dihydro-3H,7H-pyrazolo[4ʹ,3ʹ:5,6]pyrano[4,3-c][1,2]oxazole (trans-***10***)

This compound was synthesized in analogy to **5a**–**d**, except that pyrazole **9** was used as the adduct. White solid, yield 79 mg, 62%, mp 195–196 °C. IR (v_max_, cm^−1^): 2921 (CH_aliph_), 1646, 1578, 1490, 1368 (C=N, C=C, C–N, C–O–C), 978, 832, 686 (CH=CH of benzene). ^1^H NMR (700 MHz, CDCl_3_): δ 3.87 (ddd, *J* = 13.1, 12.2, 5.6 Hz, 1H, 3a-H), 4.34 (dd, *J* = 12.2, 10.6 Hz, 1H, 4-H), 4.74 (dd, *J* = 10.5, 5.6 Hz, 1H, 4-H), 5.16 (d, *J*= 13.1 Hz, 1H, 3-H), 7.31 (t, *J* = 7.4 Hz, 1H, NPh 4-H), 7.39–7.41 (m, 1H, CPh 4-H), 7.42–7.44 (m, 2H, CPh 3,5-H), 7.44–7.48 (m, 4H, CPh 2,6-H and NPh 3,5-H), 7.65–7.67 (m, 2H, NPh 2,6-H), 8.13 (s, 1H, 8-H). ^13^C NMR (176 MHz, CDCl_3_): δ 53.4 (C-3a), 70.7 (C-4), 85.0 (C-3), 96.7 (C-8a), 118.9 (NPh C-2,6), 123.6 (C-8), 126.9 (CPh C-2,6), 127.2 (NPh C-4), 129.1 (CPh C-3,5), 129.2 (CPh C-4), 129.7 (NPh C-3,5), 136.7 (CPh C-1), 139.4 (NPh C-1), 150.3 (C-8b), 162.9 (C-5a). ^15^N NMR (71 MHz, CDCl_3_): δ −177.1 (N-7), −117.4 (N-6), −31.0 (N-1). MS *m*/*z* (%): 318 ([M + H]^+^, 100). HRMS (ESI^+^) for C_19_H_15_N_3_NaO_2_ ([M + Na]^+^) requires 340.1056, found 340.1057.

### 3.10. Synthesis of 4,7-Dihydropyrazolo[4′,3′:5,6]pyrano[4,3-c]oxazole *(**6**)*

To a mixture of 1-phenyl-3-(prop-2-yn-1-yloxy)-1*H*-pyrazole-4-carbaldehyde **11** [26] (226 mg, 1 mmol) in EtOH (2 mL), sodium acetate (123 mg, 1.5 mmol) and hydroxylamine hydrochloride (104 mg, 1.5 mmol) were added portion-wise. The reaction mixture was refluxed for 15 min. After completion of the reaction as monitored by TLC, EtOH was evaporated, and the mixture was diluted with water (10 mL) and extracted with ethyl acetate (3 × 10 mL). The organic layers were combined, washed with brine, dried over Na_2_SO_4_, filtrated, and the solvent was evaporated. Obtained oxime **12** was used in the next step without further purification. To a mixture of 1-phenyl-3-(prop-2-yn-1-yloxy)-1*H*-pyrazole-4-carbaldehyde oxime (**12**) (approximately 1 mmol) in DCM (2 mL), 5.25% aq. NaOCl solution (2.35 mL, 2 mmol) was added dropwise. After stirring at room temperature for 1 h, the reaction mixture was then poured into water and extracted with ethyl acetate (3 × 10 mL). The organic layers were combined, washed with brine, dried over Na_2_SO_4_, filtrated, and the solvent was evaporated. The residue was purified by flash column chromatography (SiO_2_, eluent: ethyl acetate/*n*-hexane, 1:4, *v*/*v*) to provide the desired compound **6**. Yield 189 mg, 79%. The NMR, IR, MS, HRMS and mp data of compound **6** are given in Section 3.6.

### 3.11. General Sonogashira Reaction Procedure for the Synthesis of ***13b**,**c***

To a mixture of 1-phenyl-3-(prop-2-yn-1-yloxy)-1*H*-pyrazole-4-carbaldehyde **11** [26] (226 mg, 1 mmol) in absolute DMF (2 mL), triethyamine (0.21 mL, 1.5 mmol), appropriate (het)aryl halide (1.1 mmol), CuI (190 mg, 0.1 mmol) and Pd(PPh_3_)Cl_2_ (35 mg, 0.05 mmol) were added. The reaction mixture was stirred under an Ar atmosphere at 60 °C for 15 min. After cooling to room temperature, the reaction mixture was quenched by the addition of water (10 mL) and extracted with EtOAc (3 × 20 mL). The combined organic layers were dried over anhydrous sodium sulfate, filtered and concentrated under reduced pressure. The residue was purified by flash column chromatography (SiO_2_, eluent: ethyl acetate/*n*-hexane, 1:4, *v*/*v*) to provide the desired compounds **13b**,**c**.

#### 3.11.1. 3-{[3-(Naphthalen-1-yl)prop-2-yn-1-yl]oxy}-1-phenyl-1H-pyrazole-4-carbaldehyde (**13b**)

Compound **13b** was obtained from 11 according to the general Sonogashira reaction procedure using 1-iodonaphtalene (279 mg, 1.1 mmol). Colorless solid, yield 225 mg, 64%, mp 142–145 °C. IR (KBr, ν_max_, cm^−1^) 3127, 3096 (CH_arom_), 2917, 2826 (CH_alif_), 2250 (C≡C), 1671 (CHO), 1598, 1508, 1496, 1343, 1231 (C=C, C–N, C–O–C), 986, 770, 687 (CH=CH of benzenes). ^1^H NMR (700 MHz, CDCl_3_): δ 5.42 (s, 2H, CH_2_), 7.34–7.36 (m, 1H, Ar-H), 7.38–7.43 (m, 2H, Ar-H), 7.47–7.49 (m, 3H, NPh 3,5-H, Ar-H), 7.71–7.72 (m, 3H, NPh 2,6-H, Ar-H), 7.82–7.84 (m, 2H, Ar-H), 8.30 (s, 1H, 5-H), 8.34–8.35 (m, 1H, Ar-H), 9.93 (s, 1H, CHO). ^13^C BMR (176 MHz, CDCl_3_): δ 58.1 (CH_2_), 85.5 (C≡CNph), 88.3 (C≡CNph), 111.8 (C-4), 118.9 (NPh C-2,6), 119.9, 125.2,126.2, 126.6, 127.0, 127.5 (NPh C-4), 128.4, 129.4, 129.8 (NPh C-3,5), 129.8 (C-5), 131.0, 133.2, 133.6, 139.1 (NPh C-1), 162.7 (C-3), 183.3 (CHO). ^15^N NMR (71 MHz, CDCl_3_): δ −178.6 (N-1), −116.1 (N-2). MS *m*/*z* (%): 353 ([M + H]^+^, 100). HRMS (ESI^+^) for C_23_H_16_N_2_NaO_2_ ([M + Na]^+^) requires 375.1104, found 375.1103.

#### 3.11.2. 1-Phenyl-3-{[3-(pyridin-2-yl)prop-2-yn-1-yl]oxy}-1H-pyrazole-4-carbaldehyde (**13c**)

Compound **13c** was obtained from **11** according to the general Sonogashira reaction procedure using 2-bromopyridine (174 mg, 1.1 mmol). Colorless solid, yield 212 mg, 70%, mp 110–113 °C. IR (KBr, ν_max_, cm^−1^): 3100 (CH_arom_), 2917, 2850 (CH_aliph_), 2300 (C≡C), 1674 (CHO), 1561, 1501, 1466, 1357, 1209, 1020 (C=C, C–N), 757, 688 (CH=CH of benzene). ^1^H NMR (700 MHz, CDCl_3_): δ 5.28 (s, 2H, CH_2_), 7.23–7.25 (m, 1H, Py 3-H), 7.30–7.32 (m, 1H, Ph 4-H), 7.44–7.47 (m, 3H, Ph 3,5-H, Py 5-H), 7.63–7.65 (m, 3H, Ph 2,6-H, Py 4-H), 8.27 (s, 1H, 5-H), 8.58 (s, 1H, Py 6-H), 9.88 (s, 1H, CHO). ^13^C NMR (176 MHz, CDCl_3_): δ 57.5 (CH_2_), 83.4 (C≡CPy), 86.4 (C≡CPy), 111.5 (C-4), 118.9 (Ph C-2,6), 123.5 (Py C-5), 127.5 (Ph C-4, Py C-3), 129.7 (Ph C-3,5), 129.8 (C-5), 136.3 (Py C-4), 138.9 (Ph C-1), 142.5 (Py C-2), 150.1 (Py C-6), 162.6 (C-3), 183.3 (CHO). ^15^N NMR (71 MHz, CDCl_3_): δ −179.3 (N-1), −118.5 (N-2), pyridinyl N was not found. MS *m*/*z* (%): 304 ([M + H]^+^, 100). HRMS (ESI^+^) for C_18_H_13_N_3_NaO_2_ ([M + Na]^+^) requires 326.0900, found 326.0901.

### 3.12. General Procedure for the Synthesis of 1-Phenyl-3-(prop-2-yn-1-yloxy)-1H-pyrazole-4-carbaldehyde oximes ***14a**–**c***

To a mixture of appropriate 1-phenyl-3-(prop-2-yn-1-yloxy)-1*H*-pyrazole-4-carbaldehyde **13a**–**c** (1 mmol) in EtOH (2 mL), sodium acetate (123 mg, 1.5 mmol) and hydroxylamine hydrochloride (104 mg, 1.5 mmol) were added portion-wise. The reaction mixture was refluxed for 15 min. After completion of the reaction as monitored by TLC, EtOH was evaporated, and the mixture was diluted with water (10 mL) and extracted with ethyl acetate (3 × 10 mL). The organic layers were combined, washed with brine, dried over Na_2_SO_4_, filtrated, and the solvent was evaporated. Obtained oximes **14a**–**c** were used in the next step without further purification.

#### Data for Selected Oxime. N-[(Z)-{1-Phenyl-3-[(3-phenylprop-2-yn-1-yl)oxy]-1H-pyrazol-4-yl}methylidene]hydroxylamine (**14a**)

White solid, yield 245 mg, 77%, mp 172–174.5 °C. IR (KBr, ν_max_, cm^−1^): 3166 (OH), 3130, 3065, 3021 (CH_arom_), 2999, 2813 (CH_aliph_), 2237 (C≡C), 1642, 1600, 1565, 1504, 1489, 1401, 1352, 1218, 1174, 1056, 1009 (C=N, C=C, C–N, C–O–C), 996, 966, 931, 903, 874, 711, 682 (CH=CH of benzenes). ^1^H NMR (700 MHz, DMSO-*d*_6_): δ 5.29 (s, 2H, CH_2_), 7.27 (s, 1H, CHNOH), 7.28–7.30 (m, 1H, NPh 4-H), 7.38–7.42 (m, 3H, CPh 3-5-H), 7.47–7.50 (m, 4H, NPh 3,5-H, CPh 2,6-H), 7.82–7.83 (m, 2H, NPh 2,6-H), 8.87 (Pz 5-H), 11.65 (OH). ^13^C NMR (176 MHz, DMSO-*d*_6_): δ 57.3 (CH_2_), 84.7 (C≡CPh), 86.3 (C≡CPh), 100.6 (Pz C-4), 118.0 (NPh C-2,6), 121.5 (CPh C-1), 126.1 (NPh C-4), 128.8 (CPh C-3,5), 129.2 (CPh C-4), 129.6 (NPh C-3,5), 131.3 (Pz C-5), 131.6 (CPh C-2,6), 134.7 (CH=N–OH), 139.1 (NPh C-1), 160.6 (Pz C-3). ^15^N NMR (71 MHz, DMSO-*d*_6_): δ −184.4 (N-1), −122.6 (N-2), −18.4 (CH=N–OH). MS *m*/*z* (%): 318 ([M + H]^+^, 100). HRMS (ESI^+^) for C_19_H_16_N_3_O_2_ ([M + H]^+^) requires 318.1237, found 318.1238.

### 3.13. General Procedure for the Synthesis of 4H,7H-Pyrazolo[4′,3′:5,6]pyrano[4,3-c]oxazoles ***15a**–**c***

To a mixture of appropriate 1-phenyl-3-(prop-2-yn-1-yloxy)-1*H*-pyrazole-4-carbaldehyde oxime **14a**–**c** (approximately 1 mmol) in DCM (2 mL), 5.25% aq. NaOCl solution (2.35 mL, 2 mmol) was added dropwise. After stirring at room temperature for 1 h, the reaction mixture was quenched by the addition of water (20 mL) and extracted with DCM (3 × 20 mL). The combined organic layers were dried over anhydrous sodium sulfate, filtered and concentrated under reduced pressure. The residue was purified by flash column chromatography (SiO_2_, eluent: ethyl acetate/*n*-hexane,1:4, *v*/*v*) to provide the desired compounds **15a**–**c**.

#### 3.13.1. 3,7-Diphenyl-4H,7H-pyrazolo[4′,3′:5,6]pyrano[4,3-c]oxazole (**15a**)

White solid, yield 239 mg, 76%, mp 209–212 °C. IR (KBr, ν_max_, cm^−1^): 3050 (CH_arom_), 1640, 1602, 1526, 145 3, 1405, 1375, 1340, 1048 (C=N, C=C, C–N, C–O–C), 956, 750, 725, 682 (CH=CH of benzenes). ^1^H NMR (700 MHz, TFA-*d*): δ 6.03 (s, 2H, CH_2_), 7.57–7.65 (m, 10H, Ph-H), 8.63 (s, 1H, 8-H). ^13^C NMR (700 MHz, TFA-*d*): δ 69.7 (CH_2_), 95.6 (C-3a), 102.6 (C-8a), 122.1 (NPh C-2,6), 124.7 (C-8), 126.3, 129.2, 130.4, 130.8, 131.4, 131.9, 134.3, 148.1, 158.4 (C-3), 166.9 (C-5a). MS *m*/*z* (%): 316 ([M + H]^+^, 100). HRMS (ESI^+^) for C_19_H_13_N_3_NaO_2_ ([M + Na]^+^) requires 338.0900, found 338.0902.

#### 3.13.2. 3-(Naphthalen-1-yl)-7-phenyl-4H,7H-pyrazolo[4′,3′:5,6]pyrano[4,3-c]oxazole (**15b**)

White solid, yield 281 mg, 77%, mp 202–205 °C. IR (KBr, ν_max_, cm^−1^): 3057 (CH_arom_), 1646, 1598, 1530, 1472, 1395, 1256, 1047 (C=N, C=C, C–N, C–O–C), 947, 802, 772, 684 (CH=CH of benzenes). ^1^H NMR (700 MHz, CDCl_3_): δ 5.32 (s, 2H, CH_2_), 7.23–7.25 (m, 1H, Ph 4-H), 7.40–7.42 (m, 2H, Ph 3,5-H), 7.50–7.56 (m, 4H, Nph 2,3,6,7-H), 7.63–7.64 (m, 2H, Ph 2,6-H), 7.88–7.89 (m, 1H, Nph 5-H), 7.94–7.96 (m, 1H, Nph 4-H), 8.00–8.01 (m, 1H, Nph 8-H), 8.20 (s, 1H, 8-H). ^13^C NMR (176 MHz, CDCl_3_): δ 64.4 (CH_2_), 97.1 (C-8a), 107.7 (C-3a), 118.6 (Ph C-2,6), 122.6 (C-8), 124.3 (Nph C-1), 124.9 (Nph C-8), 125.1 (Nph C-3), 126.7 (Ph C-4), 126.8 (Nph C-6), 127.6 (Nph C-7), 127.9 (Nph C-2), 128.7 (Nph C-5), 129.6 (Ph C-3,5), 130.6 (Nph C-8a), 131.2 (Nph C-4), 133.8 (Nph C-4a), 139.5 (Ph C-1), 152.2 (C-8b), 162.5 (C-5a), 162.6 (C-3). ^15^N NMR (71 MHz, CDCl_3_): δ −179.6 (N-7), −116.4 (N-6), −23.9 (N-1). MS *m*/*z* (%): 366 ([M + H]^+^, 100). HRMS (ESI^+^) for C_23_H_15_N_3_NaO_2_ ([M + Na]^+^) requires 388.1056, found 388.1054.

#### 3.13.3. 7-Phenyl-3-(pyridin-2-yl)-4H,7H-pyrazolo[4′,3′:5,6]pyrano[4,3-c]oxazole (**15c**)

White solid, yield 234 mg, 74%, mp 241–244 °C. IR (KBr, ν_max_, cm^−1^): 3100 (CH_arom_), 1638, 1600, 1529, 1466, 1410, 1380, 1344, 1052 (C=N, C=C, C–N, C–O–C), 958, 753, 729, 685 (CH=CH of benzene). ^1^H NMR (700 MHz, CDCl_3_): δ 5.84 (s, 2H, CH_2_), 7.29–7.31 (m, 1H, Ph 4-H), 7.32–7.35 (m, 1H, Py 5-H), 7.45–7.48 (m, 2H, Ph 3,5-H), 7.69–7.70 (m, 2H, Ph 2,6-H), 7.85 (t, *J* = 7.7 Hz, 1H, Py 4-H), 7.95 (d, *J* = 7.8 Hz, 1H, Py 3-H), 8.22 (s, 1H, 8-H), 8.70 (d, *J* = 4.5 Hz, 1H, Py 6-H). ^13^C NMR (176 MHz, CDCl_3_): δ 65.6 (CH_2_), 96.5 (C-8a), 108.8 (C-3a), 118.7 (Ph C-2,6), 121.3 (Py C-3), 122.6 (C-8), 124.2 (Py C-5), 126.8 (Ph C-4), 129.8 (Ph C-3,5), 137.1 (Py C-4), 139.7 (Ph C-1), 147.2 (Py C-2), 150.2 (Py C-6), 152.7 (C-8b), 160.6 (C-3), 162.7 (C-5a). ^15^N NMR (71 MHz, CDCl_3_): δ −180.1 (N-7), −117.4 (N-6), −72.8 (pyridine N), −25.0 (N-1). MS *m*/*z* (%): 317 ([M + H]^+^, 100). HRMS (ESI^+^) for C_18_H_12_N_4_NaO_2_ ([M + Na]^+^) requires 339.0852, found 339.08555.

## 4. Conclusions

In conclusion, we have developed a convenient method for the preparation of 3a,4-dihydro-3*H*,7*H*- and 4*H*,7*H*-pyrazolo[4′,3′:5,6]pyrano[4,3-*c*][1,2]oxazoles from easily obtainable 3-(prop-2-en-1-yloxy)- or 3-(prop-2-yn-1-yloxy)-1*H*-pyrazole-4-carbaldehydes by INOC reaction of intermediate oximes. The key stage—nitrile oxide preparation from the corresponding aldoximes—was carried out by oxidation with sodium hypochlorite. The method was applied for the synthesis of pyrazolo[4′,3′:5,6]pyrano[4,3-*c*][1,2]oxazoles with various substituents in the third or seventh position. In addition, extensive NMR spectroscopic studies have been undertaken using standard and advanced methods to unambiguously determine the configuration of intermediate aldoximes, showing the predomination of the *syn*-isomer, as well as the structure of new polycyclic systems.

## Data Availability

The data presented in this study are available on request from the corresponding authors.

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
