# Peer review of "Convenient Synthesis of Pyrazolo[4′,3′:5,6]pyrano[4,3-*c*][1,2]oxazoles via Intramolecular Nitrile Oxide Cycloaddition"

_molecules, 2021, doi:10.3390/molecules26185604_

Round 1

Reviewer 1 Report

This article describes a simple and convenient synthesis of new dihydropyrazolo[4',3':5,6]pyrano[4,3-c][1,2]oxazole derivatives from 3-(prop-2-en-1-yloxy)- or 3-(prop-2-yn-1-yloxy)-1H-pyrazole-4-carbaldehyde oximes via an intramolecular nitrile oxide cycloaddition reaction. In general, the thematic and obtained results are interesting for chemists of synthetic organic field, the synthetic methodology is relatively simple and practical, and all the products are well characterized. Notably, the authors determined the configuration of intermediate aldoximes by using NOESY experimental data and analyzing the respective coupling constants with exceptional conclusions. In addition, structures of products were confirmed by detailed NMR spectroscopic experiments and HRMS analysis. In general, the manuscript is very well written and analyzed, mainly in the optimization and characterization studies; indeed,  the characterization of all the compounds obtained (they are really many) is excellent and complete. On balance, this assessment is optimistic; I believe this paper is appropriate for its publication in Molecules after minor revisions.

  1. The manuscript title would be better as 'Convenient synthesis of pyrazolo[4',3':5,6]pyrano[4,3-c][1,2]oxazoles via intramolecular nitrile oxide cycloaddition.
  2. It would be interesting to study the methodology with other N-arylpyrazoles. Anyway, the work is complete like this.
  3. There are some errors in punctuation that should be checked, mainly the absence of several commas (,).
  4. Conclusions must be improved.

Reviewer 2 Report

This manuscript by Milišiūnaitė et al. describes the synthesis of some novel pyrazolopyrano-fused isoxazoline and isoxazole derivatives by the intramolecular cycloaddition of nitrile-oxide dipoles, generated in situ from oximes, to alkene or alkyne dipolarophile moieties.

The work fits within the topic of the special issue to which it was submitted. The article is well designed, the synthetic procedures led to the desired products in moderate to high yields. The structures of the synthesised compounds were unambiguously determined by different NMR methods supplemented by theoretical calculation in some cases.  

Only some minor points are suggested for consideration:

  1. Abstract, 16th row: …oximes have been developed instead of ….has been developed
  2. Abbreviated first names should be deleted for authors cited, rows 33,36, 133, 150, 244 (e.g. Svestrupor et al, instead of T. D. Svestrupor)
  3. row 67:pyrazole-containing instead of pyrazole-moiety-containing
  4. row 75: attached to instead of attached onto
  5. Scheme 1: 4-F-C6H4, 4-Br- C6H4 throughout instead of 4-FPh, 4-BrPh
  6. row 106: ….,while the minor isomer showed…..instead of …,while in the case of the minor isomer, it only showed…
  7. Table 1. Yield (%) row, delete % after 20, 68 and 52
  8. Scheme 2: I do not understand why two configurational isomers of the same structure are indicated with different numbers, trans-10 and cis-11, please correct to trans-10 and cis-10 in the scheme and also throughout the text.
  9. rows 265-266: …..,which was easily elucidated from a 1JCH coupling constant (179.2 Hz) of the iminyl moiety.
  10. Scheme 3: renumber the structures from 11 to 15 (also in the related text)

In summary, in my opinion, the manuscript can be accepted for publication in Molecules after the above corrections.
